# Efficient Phosphorus Recovery from Municipal Wastewater Using Enhanced Biological Phosphorus Removal in an Anaerobic/Anoxic/Aerobic Membrane Bioreactor and Magnesium-Based Pellets

**DOI:** 10.3390/membranes12020210

**Published:** 2022-02-10

**Authors:** Elvis Eghombi, Hyunsik Kim, Yang-Hun Choi, Mi-Hwa Baek, Mallikarjuna N. Nadagouda, Pyung-Kyu Park, Soryong Chae

**Affiliations:** 1Department of Chemical and Environmental Engineering, University of Cincinnati, 2901 Woodside Drive, Cincinnati, OH 45221, USA; eghombielvis@yahoo.com (E.E.); kim4hk@mail.uc.edu (H.K.); 2Water Treatment Development Team, LOTTE Chemical Advanced Materials, 56 Gosan-ro, Uiwang-si 16073, Korea; yanghunchoi@lotte.net; 3Monomer R&D Division, LOTTE Chemical R&D Center, 115 Gajeongbuk-ro, Daejeon 34110, Korea; mihwabaek@lotte.net; 4Department of Mechanical and Materials Engineering, Wright State University, Dayton, OH 45435, USA; nadagouda.mallikarjuna@epa.gov; 5Department of Environmental and Energy Engineering, Yonsei University, 1 Yonseidae-gil, Wonju 26493, Korea

**Keywords:** phosphorus, municipal wastewater, anaerobic/anoxic/aerobic membrane bioreactor, enhanced biological phosphorus removal mechanism, ethanol, membrane fouling, extracellular polymeric substances

## Abstract

Municipal wastewater has been identified as a potential source of natural phosphorus (P) that is projected to become depleted in a few decades based on current exploitation rates. This paper focuses on combining a bench-scale anaerobic/anoxic/aerobic membrane bioreactor (MBR) and magnesium carbonate (MgCO_3_)-based pellets to effectively recover P from municipal wastewater. Ethanol was introduced into the anoxic zone of the MBR system as an external carbon source to improve P release via the enhanced biological phosphorus removal (EBPR) mechanism, making it available for adsorption by the continuous-flow MgCO_3_ pellet column. An increase in the concentration of P in the MBR effluent led to an increase in the P adsorption capacity of the MgCO_3_ pellets. As a result, the anaerobic/anoxic/aerobic MBR system, combined with a MgCO_3_ pellet column and ethanol, achieved 91.6% P recovery from municipal wastewater, resulting in a maximum P adsorption capacity of 12.8 mg P/g MgCO_3_ through the continuous-flow MgCO_3_ pellet column. Although the introduction of ethanol into the anoxic zone was instrumental in releasing P through the EBPR, it could potentially increase membrane fouling by increasing the concentration of extracellular polymeric substances (EPSs) in the anoxic zone.

## 1. Introduction 

Natural phosphorus (P) reserves will be depleted in a few decades if the phosphorus fertilizer demand increases at 3% per year [1]. Municipal wastewater has been identified as an alternative P source that potentially reduces natural P extraction from phosphate rocks [2]. It has been estimated that 15–20% of the world’s phosphorus demand could be satisfied by its recovery from municipal wastewater [1,3,4]. A recent study showed that humans discharge about 3.7 Mt of P into wastewater, making its recovery from wastewater a desirable alternative capable of providing sustainable phosphorus supplies that could supplement natural phosphorus [4].

This essential element (i.e., P) in sewage can be recovered through chemical precipitation, biological processes, physical adsorption, sewage sludge, wetland plants, and wastewater irrigation. Chemical precipitation, physical adsorption, and biological removal using the enhanced biological phosphorus removal (EBPR) mechanism are the most widely used methods [5].

Struvite (MgNH_4_PO_4_·6H_2_O), or calcium phosphate precipitation, is also regarded as one of the most promising phosphorus recovery processes since the resultant products can be directly applied in agriculture as a fertilizer or accepted readily by the phosphorus manufacturing industry [1,6,7]. However, this recovery method with chemicals requires wastewater with a high phosphorus concentration to be feasible, which is not the case with municipal wastewater that is naturally diluted with a phosphorus concentration of less than 10 mg/L [8].

Adsorptive materials such as modified iron oxide iron oxide, calcined waste eggshells, and magnesium-modified corn biochar have also been proven to effectively remove and recover phosphate from aqueous suspensions. However, a major setback with this technology is the “bottleneck” phenomenon where the adsorbent is no longer applicable after saturation [9,10,11,12]. Therefore, this necessitates fabricating an excellent adsorbent material with highly recyclable properties that can be used as an alternative in P recovery from wastewater.

Magnesium carbonate (MgCO_3_)-based materials, especially beads and pellets, are adsorptive materials that have exhibited excellent capabilities for effective and sustainable recovery of P from wastewater. The effectiveness of the pellets is attributed to MgCO_3_’s chemical and physical stability in water and its environmentally friendly properties that guarantee its application in agriculture as a slow-release material [13,14].

The biological removal method using the EBPR mechanism is a relatively inexpensive and ecologically friendly process used in phosphorus recovery even though its stability and reliability are hard to attain [15]. Biological phosphorus removal via the EBPR process can accumulate up to 90% of P in sludge [16]. Ultimately, phosphorus removal is achieved by withdrawing excess sludge containing the accumulated poly-P from the engineered system [17].

The effectiveness of the EBPR process is determined by the type and amount of carbon sources available in the wastewater under treatment [18]. Among various carbon sources, ethanol is preferably used by most wastewater treatment plants (WWTPs) in the EBPR process because it is cheap and sustainable compared to other carbon sources [19]. 

Although the EBPR process is an environmentally sustainable method for phosphorus recovery from municipal wastewater, it has some limitations in that it is unstable and unreliable [15]. In addition, phosphorus removal is achieved by discharging the phosphorus-rich excess sludge with heavy metals and pathogens [20], resulting in additional disposal costs since the recovered phosphorus is not separated from the sludge [21].

Membrane bioreactors (MBRs) have been widely used in the treatment of municipal and industrial wastewater as membrane separation technology is increasingly becoming an innovation in biological wastewater treatment [22,23,24]. Because of the high quality (particle and bacteria free) of the resultant effluent from MBRs, the effluent can be directly discharged into the environment or used directly for non-potable applications such as irrigation and industrial applications.

Although MBRs are a stable, reliable, and sustainable wastewater treatment method that achieve excellent solid–liquid separation, they still cannot recover the phosphorus released in the liquid stream. Therefore, another technology that could be operated with an MBR to recover P from the liquid stream discharged is needed. 

In our previous studies, granular adsorptive pellets were fabricated by combining MgCO_3_ with varying amounts of cellulose binder to remove phosphate from water. In a batch experiment, a maximum adsorption capacity of 96.4 mg of P per gram of MgCO_3_ was achieved from a synthetic orthophosphate solution (initial P concentration = 160 mg/L) for 25 days, following a pseudo-second-order kinetics model [14]. 

The following study was tested using real municipal wastewater in combination with MBR. The MBR effluent was directly fed to a continuous flow column packed with MgCO_3_ pellets at various flow rates (i.e., 10 L/d to 2.5 L/d). Under the optimum conditions, 73.1% of phosphorus was recovered from the MBR effluent, but the phosphorus’s adsorption capacity using MgCO_3_ pellets was limited to 0.47 mg of phosphorus per gram of MgCO_3_ [25]. By switching the feed water from a synthetic solution to natural municipal wastewater, there was a dramatic change in the P adsorption capacity of MgCO_3_ from 96.4 mg of P per gram of MgCO_3_ to 0.47 mg of P per gram of MgCO_3_. 

The low adsorption capacity obtained in the previous study using real municipal wastewater requires a follow-up study to improve the phosphorus recovery efficiency. Therefore, this study aimed to improve P release from an anaerobic/anoxic/aerobic MBR via the EBPR mechanism by adding an external carbon source (i.e., ethanol) to enhance the adsorption capacity of P using MgCO_3_ pellets. 

The MBR technology can efficiently remove particles, heavy metals, and pathogens from a mixed liquor that traditional biological processes cannot usually achieve [26,27,28,29]. Then, the particle- and pathogen-free MBR effluent was introduced to a continuous flow column reactor packed with MgCO_3_ pellets to recover the phosphorus. The effects of ethanol injection on P recovery and membrane fouling were determined. 

## 2. Materials and Methods 

### 2.1. Characterization of Municipal Wastewater

Municipal wastewater used for this experiment was collected from the Muddy Creek WWTP (Cincinnati, OH, USA) after primary settling. The temperature and pH were measured using a bench-top pH meter (Thermo Scientific, Waltham, MA, USA). 

Key wastewater parameters such as chemical oxygen demand (COD), total phosphorus (TP), orthophosphate, total Kjeldahl nitrogen (TKN), ammonia, nitrates, nitrites, total suspended solids (TSS), mixed liquor suspended solids (MLSS), volatile suspended solids (VSS), and fecal coliform were analyzed at least three times per week using an ultraviolet/visible (UV/Vis) spectrophotometer (DR6000, HACH, Loveland, CO, USA) according to standard methods [30]. 

The detection limits of the key water quality parameters were: COD = 1–60 mg/L (ultra-low range) and 20–1500 mg/L (high range), phosphorus = 0.15–4.50 mg/L as PO_4_^3−^, Total Kjeldahl nitrogen (TKN) = 0–16 mg/L as N, NH_3_-N = 0.02–2.50 mg/L (low range) and 0.4–50.0 mg/L (high range), NO_3_-N = 0.23–13.50 mg/L, and NO_2_-N = 0.015–0.6 mg/L.

The primary effluent from Muddy Creek WWTP had an average concentration of total COD and TP of 217 mg/L and 4.1 mg/L, respectively. The characteristics of the primary effluent during the experimental period are shown in Table 1. 

### 2.2. Bench-Scale Anaerobic/Anoxic/Aerobic MBR System

A bench-scale anaerobic/anoxic/aerobic MBR has three compartments: an anaerobic zone (1.0 L), an anoxic zone (2.4 L), and an aerobic zone (8.0 L), with a total MBR operating volume of 11.4 L (Figure 1). The MBR was inoculated with sludge from the Muddy Creek WWTP (Cincinnati, OH, USA). The primary effluent from the Muddy Creek WWTP and MLSS recycled from the aerobic zone were introduced to the top of the anaerobic zone (oxidation–reduction potential (ORP) < −450 mV) of the MBR to deplete the dissolved oxygen (DO) in the MLSS and moved to the anoxic zone. Then, the MLSSs overflowed from the top of the anaerobic zone to the anoxic zone (ORP = −300 ~ −150 mV) to the aerobic zone (ORP > 200 mV). Complete mixing in the anoxic zone was obtained using a low-speed mixer and an impeller. Under alternating anaerobic/anoxic and aerobic conditions, the MBR achieved the EBPR mechanism, biological nitrification, and denitrification. Ethanol introduced into the anoxic zone enhanced the EBPR mechanism and denitrification processes.

A hollow fiber polyvinylidene fluoride (PVDF) membrane module (with a nominal pore size of 0.03 μm; LOTTE Chemical, Daegu, South Korea) was submerged in the aerobic zone (Phases 1–5) or the anoxic zone (Phases 6 and 7). Detailed information on the PVDF membrane can be found in our previous study [25]. 

A disk-type diffuser was installed in the aerobic zone at the bottom to provide air bubbles for the oxidation of organics and ammonia and to reduce membrane fouling. The DO concentration in the aerobic zone ranged between 3 and 5 mg/L. In all the phases, the MBR effluent was withdrawn using a vacuum pump through the PVDF membrane module at 15 L/m^2^/h (LMH). The MLSS were internally recycled back to the anaerobic zone at 300% of the feed flow rate from the aerobic tank.

The MBR was operated for various hydraulic retention times (HRTs) in seven different phases (Table 2). In Phases 1–3, the MBR was operated at an 16 h HRT (Phase 1), 12 h HRT (Phase 2), and 8 h (Phase 3) HRT by altering the feed flow rate without ethanol injection or a MgCO_3_ pellet column and lasted for 90 days. In Phase 4, the MBR was operated at an 8 h HRT with ethanol for 30 days (100 mg/L as the COD ethanol solution was directly injected into the anoxic zone to increase the soluble COD/NH_3_-N ratio of the influent from 2.5 to 7.5 in the anoxic zone for EBPR and biological denitrification). 

The MgCO_3_ pellet column was connected to the MBR system in Phases 5–7 and operated to recover P from the MBR effluent for an 8 h HRT for 30 days in each phase. The removal of particles and microorganisms was achieved by membrane filtration in the MBR, while the MgCO_3_ pellet column achieved further P recovery through physical adsorption. After every experimental phase, the membrane module was replaced with a new one. Phases 5 and 6 lasted 30 days each, while Phase 7 lasted only 15 days.

### 2.3. A Continuous Flow Column Reactor with MgCO_3_ Pellets

The MgCO_3_ pellet column designed for P recovery from the MBR effluent was introduced in Phases 5–7 (Table 2). The MgCO_3_ pellets used in this research experiment were loaded into a cylindrical column, having a length of 75 cm and a diameter of 3.5 cm, with a total column volume of 0.72 L. The column was packed with approximately 110 g of MgCO_3_ pellets. The MgCO_3_ pellets were lined at the top and bottom with 80 g of gravel to remove any solid particles that might have made their way into the column. The entire packing density of the column was 153.1 g/L and was operated at room temperature (20–25 °C). 

In Phase 5 (i.e., 100% of the MBR effluent was withdrawn from the PVDF membrane submerged in the aerobic zone), we introduced the MBR effluent into the MgCO_3_ pellet column for P recovery. In Phase 6, 50% of the MBR effluent was withdrawn from the PVDF membrane submerged in the anoxic reactor for P recovery using the MgCO_3_ pellet column. The other 50% was withdrawn from the PVDF membrane submerged in the aerobic zone but discarded without P recovery. In Phase 7, 100% of the MBR effluent was withdrawn from the PVDF membrane submerged in the anoxic reactor and then passed through the MgCO_3_ pellet column. The amount of P recovered and the removal efficiency of P by the MgCO_3_ pellet column were determined using Equations (1) and (2), respectively:(1)$$\mathrm{Adsorption}\;\mathrm{capacity}\;(\mathrm{mg}\;P/g\;{\mathrm{MgCO}}_{3}-\mathrm{based}\;\mathrm{pellets})=\frac{\sum \left(C_{\mathrm{in}}-C_{\mathrm{out}}\right)Q}{W}$$
where $C_{\mathrm{in}}$ and $C_{\mathrm{out}}$ are the daily average concentration (mg/L) of P in the influent and effluent water in the column, respectively. W is the weight (g) of the MgCO_3_ pellets, and Q is the flow rate in the column (L/d).
(2)$$\mathrm{Removal}\;\mathrm{efficiency}=\left[\left(C_{\mathrm{in}}-C_{\mathrm{out}}/C_{\mathrm{in}}\right)\right]\;\times \;100$$
where $C_{\mathrm{in}}$ and $C_{\mathrm{out}}$ are the daily average concentration of various constituents in the influent and effluent (mg/L), respectively.

### 2.4. Characterization of Membrane Biofouling

The samples were collected on the surfaces of fouled membranes (submerged either in the anoxic or aerobic zones), and extracellular polymeric substances (EPSs) were determined using the colorimetric analysis method based on their applicability to characterize membrane biofouling. Two major EPS fractions of interest (i.e., protein and carbohydrate EPSs) were determined using the modified Lowry method [31] and the phenol sulfuric acid method [32], respectively. 

The applied methods were investigated in terms of their sensitivity to the selected standard compounds. The standard for proteins and polysaccharides was analyzed for their interference in all the applied colorimetric methods. The EPSs were dissolved in 0.02 M sodium hydroxide and analyzed at 200 mg/L and 100 mg/L concentrations. The standard lines were prepared in a concentration range from 5 to 100 mg/L. The absorbance of the EPS samples was then compared to that of the prepared standards. Cross-interference of the standard compounds was tested at 50 mg/L, 100 mg/L, and 1000 mg/L [33].

## 3. Results and Discussion 

### 3.1. Performance of the Anaerobic/Anoxic/Aerobic MBR System

The MBR system was operated in seven different phases according to the operations and design conditions listed in Table 2. Phases 1–3 were operated without an external carbon source, which was later introduced into the system during Phase 4 and used throughout the remaining three phases. Although the HRT decreased from 16 h (Phase 1) to 8 h (Phase 3), the COD concentration in the MBR effluent ranged between 2.5 and 4.5 mg/L (>97% removal). The concentrations of total nitrogen (TN) and TP ranged between 13.8 and 19.9 mg/L and between 1.7 and 3.7 mg/L, respectively. The MBR achieved a >99.9% removal efficiency for both TSS and fecal coliforms (5 log removal) in these phases (Table 3).

The high removal efficiencies for nutrients (i.e., TN and TP) achieved in Phase 4, as compared to the other three phases, was because the ethanol injection improved denitrification efficiency by serving as a substrate supplement for heterotrophic and facultative bacteria present in the system. Ethanol also improved the EBPR mechanism by encouraging the growth of PAOs, which have an unusual ability to release and uptake P from the system [18]. The high removal efficiencies in Phase 4 led to a better effluent quality than that in the first three phases (Table 4).

After introducing ethanol into the anoxic MBR zone as an external carbon source, the MBR effluent was introduced to a continuous flow column packed with MgCO_3_ pellets designed for P adsorption in Phases 5–7. In Phase 5, 100% of the water flux extracted from the aerobic reactor was introduced into the MgCO_3_ column at a flow rate of 15 LMH for 30 days.

The COD and TN concentrations in the MgCO_3_ column effluent were marinated at 3.0 mg/L. The TP concentration was also reduced from 1.6 mg/L in the MBR effluent to 0.4 mg/L in the MgCO_3_ column effluent, improving the TP removal efficiency significantly, from 38.5% when using the MBR to 84.6% when using the entire system (i.e., MBR + MgCO_3_ column). Both the MBR and MgCO_3_ pellet column showed similar removal efficiencies of >99.9% for TSS and fecal coliforms. The higher removal efficiencies were observed for COD, TN, and TP due to the further removal of organic matter, ammonia, and P by the MgCO_3_ pellet column during P adsorption (Table 5).

However, in Phase 6, only 50% of the water flux (i.e., 7.5 LMH) was withdrawn from the membrane submerged in the anoxic MBR zone, then introduced to the MgCO_3_ column for P recovery. The other 50% was withdrawn from the aerobic MBR zone and discarded without P recovery. As shown in Table 6, the COD concentration in the aerobic MBR effluent was 4.2 mg/L with a removal efficiency of 96.6%. A higher COD concentration of 7.6 mg/L was observed in the anoxic MBR effluent and the MgCO_3_ system’s effluent with a reduced removal efficiency of 93.8% compared to what was observed in the aerobic MBR effluent. The low COD concentration in the anoxic MBR effluent and MgCO_3_ system’s effluent compared to that in the aerobic MBR effluent can be attributed to withdrawal from the anoxic MBR zone where some organic matter was still unoxidized by microorganisms. 

The concentration of TP in the aerobic MBR effluent was 1.2 mg/L with a removal efficiency of 29.4%. In the anoxic MBR effluent and MgCO_3_ effluent, the TP concentration dropped to 0.6 mg/L with a removal efficiency of 64.7%. The high TP removal efficiency observed by the MgCO_3_ pellets in this phase can be attributed to the EPBR mechanism in the anoxic MBR zone (where withdrawal took place) that released P and made it available for adsorption by the MgCO_3_ pellet column. 

A high TN concentration of 3.1 mg/L and a low removal efficiency of 67.7% were observed in the aerobic MBR effluent because of nitrification in this reactor where oxygen was present. On the contrary, a low TN concentration of 0.4 mg/L and a high removal efficiency of 95.8% were observed in the anoxic MBR effluent and MgCO_3_ effluent due to direct withdrawal from the anoxic MBR zone where denitrification was taking place because of the absence of oxygen. 

This high performance by the anoxic MBR and MgCO_3_ effluent system in terms of TN removal was also due to further removal of ammonia by the MgCO_3_ pellet column (Table 6). As reported in our previous study, the presence of phosphate and ammonia in an equal molar ratio and the MgCO_3_ pellets could lead to the formation of struvite (i.e., Mg^2+^ + NH_4_^+^ + PO_4_^3−^ + 6H_2_O → NH_4_MgPO_4_∙6H_2_O), resulting in the removal of ammonia from the MBR effluent [25].

Phase 7 was the last phase of the experiment. During this phase, 100% of the water flux was withdrawn from the anoxic MBR zone, then introduced into the MgCO_3_ column for P recovery. This lasted only 15 days due to operational challenges caused by high rates of membrane fouling (this will be discussed in Section 3.3 below). 

As shown in Table 7, the COD concentrations in the anoxic MBR effluent and MgCO_3_ effluent were maintained at 8.5 mg/L (90.9% removal) and 7.7 mg/L (91.8% removal), respectively. The TN concentration was 4.3 mg/L in the anoxic MBR effluent and 2.6 mg/L in the MgCO_3_ effluent, achieving a removal efficiency of 51.7 % and 70.8%, respectively. 

The highest TP removal (91.6%) throughout the entire experimental operation was achieved in Phase 7. The comparatively (compared to Phases 1–6) high TP recovered in this phase can be attributed to high concentrations of soluble phosphorus in the MBR effluent that was released via the enhanced EBPR mechanism by ethanol (Figure 2).

In this phase, high removal efficiencies were observed for COD, TN, and TP due to the further removal of organic matter, N, and P by the MgCO_3_ pellet column. Over 99.9% removal was achieved for TSS and fecal coliforms by both the MBR and the MgCO_3_ systems (Table 7). Based on the results obtained, it can be inferred that the high P concentration in the MBR effluent increased the adsorption capacity of the MgCO_3_.

Compared to Phases 1–4, Phases 5–7 achieved relatively high removal efficiencies for TSS, COD, TN, and TP compared to those in the previous phases. The overall best system performance was observed in Phase 5 with an overall removal efficiency of >99.99% for TSS, 97.8% for COD, 76.3% for TN, and 84.6% for TP (Figure 3). Based on these results, it can be concluded that Phase 5 is the most favorable design and operating condition (i.e., 100% of the water flux extracted from the aerobic reactor was introduced into the MgCO_3_ column) for the effective removal of TSS, organic matter, and nutrients. 

### 3.2. Effective Recovery of Phosphorus from the MBR Effluent 

P recovery was achieved via the adsorption mechanism of the MgCO_3_ pellet column designed for this purpose. The capability of the column to recover P was studied in Phases 5–7 of the experiment under various operating conditions, as stated in Table 1. The P recovery capacity of the MgCO_3_-based pellets was determined using Equation (1) and is summarized in Table 8. 

It was observed that the P concentration in the MBR effluent influenced the adsorption capacity of the MgCO_3_ pellets. In other words, the adsorption capacity of the pellets had a positive correlation with the P concentration in the column’s influent. 

The higher the P concentration in the column’s influent, the higher the adsorption capacity of the pellets. This could explain why the highest P adsorption of 12.8 mg P/g MgCO_3_ recorded by the column was observed in Phase 7 (i.e., 100% water from the anoxic zone in the MBR). A somewhat high adsorption capacity of 10.2 mg P/g MgCO_3_ was also observed in Phase 5 (i.e., 100% water from the aerobic MBR zone), and a very low adsorption capacity of 2.6 mg P/g MgCO_3_ was observed in Phase 6 due to reduced water flux and a low P concentration in the MBR effluent compared to Phase 7 (Table 6 and Table 7).

### 3.3. Characteristics of Membrane Fouling under Different Redox Conditions

The impact of microorganisms on membrane biofouling in MBRs is inevitable [34,35,36]. As demonstrated by the transmembrane pressure (TMP), the membrane fouling rate began to rise at a slow pace, between a TMP of 0.14 and 0.19 kPa/day in Phases 1–3 (Figure 4). The moderate rate of increase over time was due to the low concentrations of EPS (<5 mg per membrane surface area, cm^2^) on the membrane surfaces in the MBR during the operating periods (data not shown). However, the dynamics changed in Phases 4–7 as an external carbon source (i.e., ethanol) was introduced into the MBR system. At this point, the rate of increase of TMP rose to about 0.45 kPa/day for Phases 4 and 5 (Figure 5). This rapid increase can be attributed to a rise in biomass concentration in the reactor, as ethanol encouraged the growth of microorganisms in the MBR. As the biomass population continued to increase with time, they secreted high concentrations of EPS into the MBR system, which caused an increase in fluid viscosity and membrane resistance, consequently leading to an increase in TMP and membrane fouling (Figure 6 and Figure 7).

The behavior of membrane fouling was different in Phases 6–7 because of a difference in the oxidation–reduction conditions under which the PVDF membrane was immersed. In the anoxic zone (Phase 6), the rate of TMP increase was about three times higher than what was observed in the aerobic MBR zone (Figure 6). 

In Phase 7, the rate of membrane fouling was alarming. The TMP’s rate of increase rose to about 1.8 kPa/day, and the operation lasted only 15 days. This increase in the membrane fouling rate in the anoxic zone was mainly due to the lack of the sufficient shear stress that can keep particles away from the membrane surface. 

In addition, high concentrations of protein and carbohydrate EPS secreted by the biomass growth could be stimulated by ethanol and adversely affected membrane filterability (Figure 7). The introduction of the external carbon source may have led to an increase in EPS concentration, which increased the rate of membrane biofouling. 

The average concentrations of carbohydrate and protein EPS fractions on the PVDF membrane were 18 mg (Phase 5)–62 mg (Phase 7) per membrane surface area (cm^2^) and 26 mg (Phase 5)–84 mg (Phase 7) per membrane surface area (cm^2^), respectively (Figure 7).

## 4. Conclusions 

The anaerobic/anoxic/aerobic MBR operated with ethanol and a continuous-flow MgCO_3_ pellet column effectively enhanced the release and recovery of phosphorus from real municipal wastewater. 

An increase in the concentration of P in the MBR effluent via the EBPR mechanism led to an increase in the adsorption capacity of the MgCO_3_ pellet column. In Phase 7, the MBR and MgCO_3_ system achieved a maximum P recovery efficiency of 91.6% and a maximum adsorption capacity of 12.8 mg P/g MgCO_3._ However, this operating condition was met with operational challenges due to severe membrane fouling within 15 days. 

The system’s overall performance was best in Phase 5 (i.e., 100% of the MBR effluent was withdrawn from the PVDF membrane submerged in the aerobic zone) of the experiment with a removal efficiency of >99.9% for TSS, >99.999% for fecal coliforms, 97.8% for COD, 76.3% for TN, and 84.6% for TP. 

This is also the ideal phase in which this system could be operated with minimal operational challenges related to membrane fouling. The MBR and MgCO_3_ system produced a high-quality effluent that was void of particles and pathogens, with low nutrient concentrations (<3.0 mg/L for TN and <0.4 mg/L for TP) in the final effluent. 

Ethanol, introduced as an external carbon source, effectively improved P release and denitrification efficiency in the MBR. The MgCO_3_ pellets were effective in the recovery of P and the removal nitrogen and organic matter after the MBR. 

## Figures and Tables

**Figure 1 membranes-12-00210-f001:**
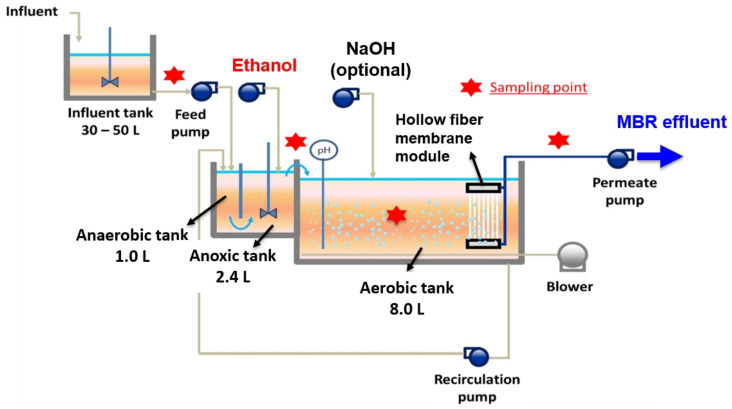
Schematic diagram of a bench-scale anaerobic/anoxic/aerobic MBR for simultaneous removal of organic matter, nitrogen, and phosphorus.

**Figure 2 membranes-12-00210-f002:**
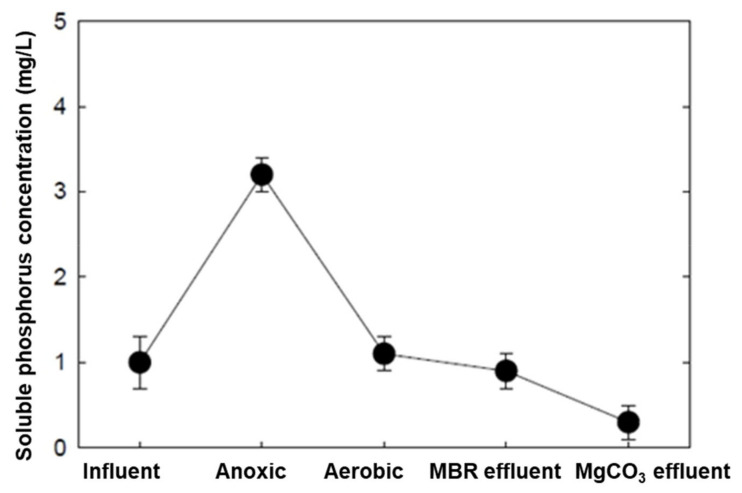
The enhanced EBPR mechanism was introduced in the MBR system by adding ethanol as an external carbon source (Phase 7).

**Figure 3 membranes-12-00210-f003:**
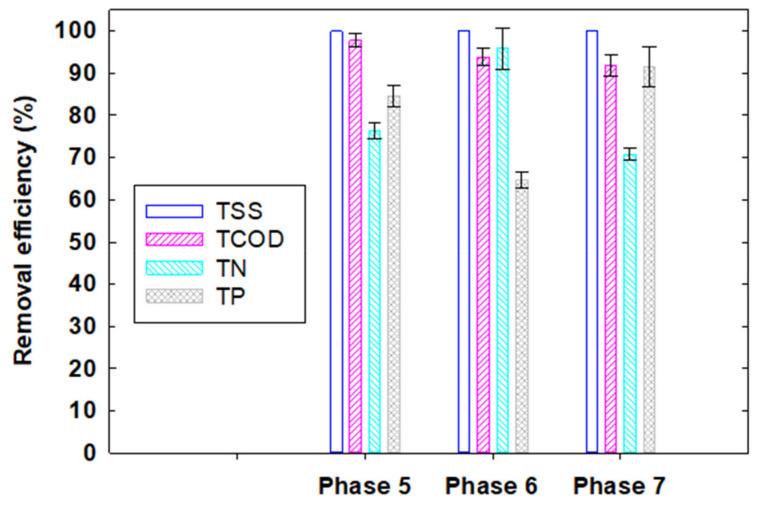
Removal efficiencies of TSS, TCOD, TN, and TP in Phases 5–7 by the aerobic MBR and MgCO_3_ column system.

**Figure 4 membranes-12-00210-f004:**
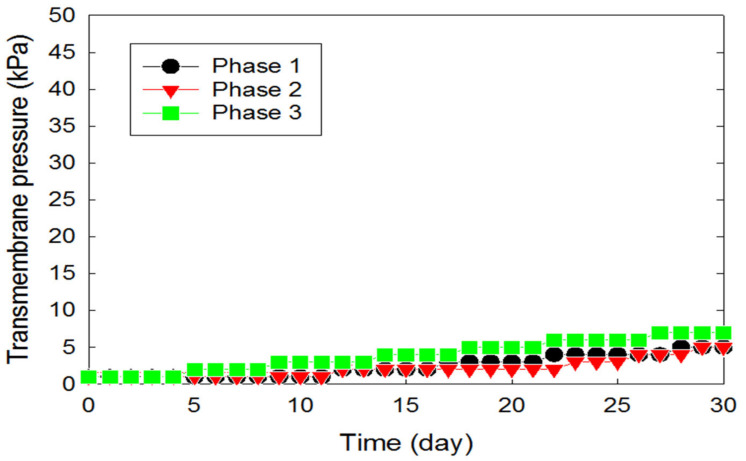
Increase in transmembrane pressure of the PVDF membrane submerged in the aerobic MBR zone as a function of time in Phase 1 (HRT = 16 h), Phase 2 (HRT = 12 h), and Phase 3 (HRT = 8 h) without ethanol.

**Figure 5 membranes-12-00210-f005:**
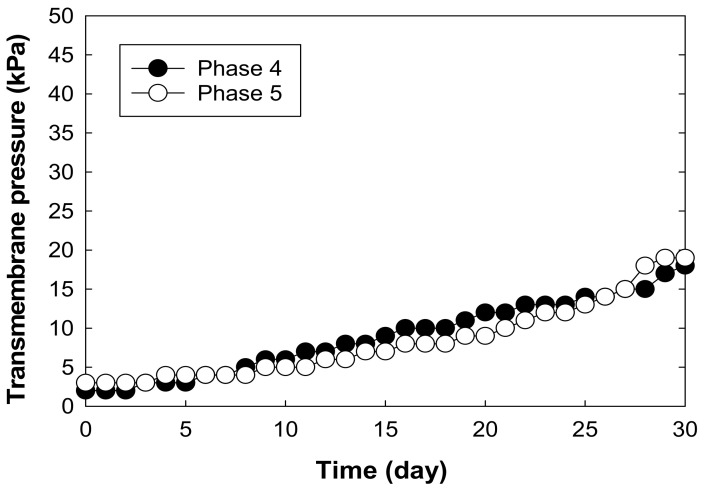
Increase in transmembrane pressure of the PVDF membrane submerged in the aerobic MBR zone as a function of time after the introduction of ethanol for an 8-h HRT in Phases 4 and 5.

**Figure 6 membranes-12-00210-f006:**
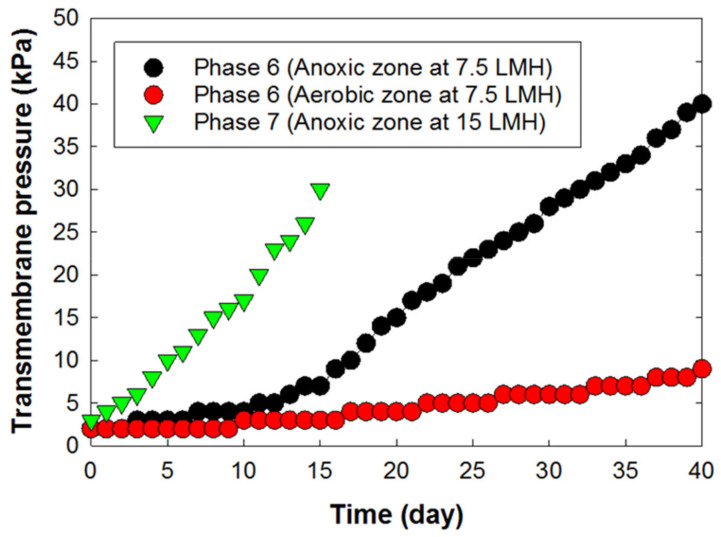
Increase in transmembrane pressure of the PVDF membrane submerged in the anoxic zones of the MBR as a function of time after introducing ethanol carbon source for an 8-h HRT in Phases 6 and 7.

**Figure 7 membranes-12-00210-f007:**
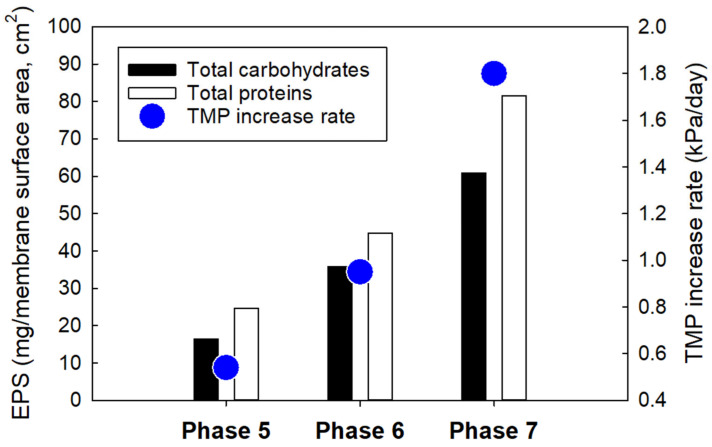
Correlation between EPS contents and transmembrane pressure of the PVDF membrane submerged in the anoxic or aerobic zones of the MBR as a function of time after introducing ethanol for an 8-h HRT in Phases 5–7.

**Table 1 membranes-12-00210-t001:** Characteristics of raw municipal wastewater and primary effluent from the Muddy Creek treatment plant (Cincinnati, OH, USA) used in this study (June 2020–February 2021).

Parameter	Raw Wastewater	Primary Effluent
Number of samples	98	98
Total suspended solids (TSS), mg/L	1015 ± 385	635 ± 317
Total chemical oxygen demand (TCOD), mg/L	261.2 ± 125.3	217 ± 123.8
Soluble chemical oxygen demand (SCOD), mg/L	61 ± 45	50 ± 31
Total Kjeldahl nitrogen (TKN), mg/L	19.2 ± 17.8	18.1 ± 16.9
NH_3_-N, mg/L	17.1 ± 8.2	16.9 ± 7.3
Total phosphorus (TP), mg/L	5.6 ± 3.3	4.1 ± 3.5
Orthophosphate, mg/L	2.2 ± 1.2	2.1 ± 1.2
Fecal coliform, CFU/100 mL	17,690 ± 7600	10,500 ± 5680

**Table 2 membranes-12-00210-t002:** Operating conditions of the reactor (Phases 1–7).

Phase	HRT (hr)	Internal Recycle Rate	Flux (LMH)	Ethanol Injection	Membrane Location	P Recovery	Period (days)
1	16	300% of Q (feed flow rate)	15	Not applicable (Soluble COD/NH_3_-N = 2.5)	Aerobic	No	30
2	12		15		Aerobic	No	30
3	8		15		Aerobic	No	30
4	8		15	100 mg/L as COD (Soluble COD/NH_3_-N = 7.5)	Aerobic	No	30
5	8		15		Aerobic	Using MgCO_3_ pellets	30
6	8		7.5		Anoxic/Aerobic	Using MgCO_3_ pellets	30
7	8		15		Anoxic	Using MgCO_3_ pellets	15

**Table 3 membranes-12-00210-t003:** Water quality of the MBR effluent (Phases 1–4).

Condition/Parameter	Phase 1	Phase 2	Phase 3	Phase 4
HRT	16	12	8	8
External carbon (ethanol) as COD (mg/L)	0	0	0	100
Number of measurements	12	12	12	12
TSS (mg/L)	ND	ND	ND	ND
TCOD (mg/L)	2.5 ± 0.8	3.9 ± 0.5	4.5 ± 1.0	5.2 ± 2.6
TN (mg/L) =TKN + NO3-N	13.8 ± 0.2	18.4 ± 0.5	19.9 ± 0.5	3.5 ± 0.4
TP (mg/L)	1.7 ± 0.3	3.3 ± 0.3	3.7 ± 0.3	2.0 ± 0.3
Fecal coliform (CFU/100 mL)	ND	ND	ND	ND

ND: not detected, CFU: colony-forming unit.

**Table 4 membranes-12-00210-t004:** Overall performance of the MBR system (Phases 1–4).

Parameter	Phase 1	Phase 2	Phase 3	Phase 4
Number of measurements	12	12	12	12
TSS (Re. %)	>99.9	>99.9	>99.9	>99.9
TCOD (Re. %)	97.4	98.1	98.2	97.7
TN (Re. %) = TKN + NO3-N	20.7	26.1	29.2	80.1
TP (Re. %)	5.6	5.7	7.5	25.9
Fecal coliform (Re. %)	>99.999	>99.999	>99.999	>99.999

**Table 5 membranes-12-00210-t005:** Performance of the MBR system and MgCO_3_ column in Phase 5.

Parameter	MBR Effluent			MBR + MgCO_3_ Effluent	
	In	Out	Re (%)	Out	Re (%)
TSS (mg/L)	585	ND	>99.9	ND	>99.9
TCOD (mg/L)	137.0	3.4	97.5	3.0	97.8
Soluble COD (mg/L)	51.8	-		-	
NH_3_-N (mg/L)	13.7	0.1	99.2	ND	>99.9
NO_2_-N and NO_3_-N (mg/L)	0.2	3.3	-	3.3	-
TN (NH_3_-N + NO_2_-N + NO_3_-N)	13.9	3.4	75.5	3.3	76.3
TP (mg/L)	2.6	1.6	38.5	0.4	84.6
Orthophosphate (mg/L)	1.7	1.6		0.4	
Fecal coliform (CFU/100 mL)	2100	ND	>99.999	ND	>99.999

ND: not detected, CFU: colony-forming unit.

**Table 6 membranes-12-00210-t006:** Performance of the MBR system and MgCO_3_ column in Phase 6.

Parameter	MBR Effluent from the Aerobic Zone (50% of Q)			MBR Effluent from Anoxic Zone (50% of Q) + MgCO_3_ Effluent		
	In	Out	Re (%)	In	Out	Re (%)
TSS (mg/L)	440	ND	>99.9	440	ND	>99.9
TCOD (mg/L)	122.2	4.2	96.6	122.2	7.6	93.8
Soluble COD (mg/L)	38.6	-		38.6	-	
NH_3_-N (mg/L)	8.5	0.1	98.8	8.5	0.1	98.8
NO_2_-N and NO_3_-N (mg/L)	1.1	3.0	-	1.1	0.3	-
TN (mg/L)	9.6	3.1	67.7	9.6	0.4	95.8
TP (mg/L)	1.7	1.2	29.4	1.7	0.6	64.7
Orthophosphate (mg/L)	1.1	1.2		1.1	0.6	

ND: not detected.

**Table 7 membranes-12-00210-t007:** Performance of the MBR system and MgCO_3_ column in Phase 7.

Parameter	MBR Effluent from the Anoxic Zone			MBR + MgCO_3_ Effluent		
	In	Out	Re (%)	In	Out	Re (%)
TSS (mg/L)	530	ND	>99.9	ND	ND	>99.9
TCOD (mg/L)	93.6	8.5	90.9	8.5	7.7	91.8
Soluble COD (mg/L)	43.2	-			-	
NH_3_-N (mg/L)	7.9	1.9	75.9	1.9	0.4	94.9
NO_2_-N and NO_3_-N (mg/L)	1.0	2.4	-	2.4	2.2	-
TN (mg/L)	8.9	4.3	51.7	4.3	2.6	70.8
TP (mg/L)	2.4	3.2	-	3.2	0.2	91.6
Orthophosphate (mg/L)	1.0	3.2			0.2	
Fecal coliform (CFU/100 mL)	2800	ND	>99.999	ND	ND	>99.999

ND: not detected, CFU: colony-forming unit.

**Table 8 membranes-12-00210-t008:** Recovery of phosphorus using MgCO_3_ pellets from the MBR system.

Phase	Water Flux Through the PVDF Membrane (LMH)	Operation Period (day)	Adsorption Capacity (mg P/g MgCO_3_)
5	15	30	10.2
6	7.5	30	2.6
7	15	15	12.8

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
