# Peer review of "Efficient Phosphorus Recovery from Municipal Wastewater Using Enhanced Biological Phosphorus Removal in an Anaerobic/Anoxic/Aerobic Membrane Bioreactor and Magnesium-Based Pellets"

_membranes, 2022, doi:10.3390/membranes12020210_

Round 1

Reviewer 1 Report

membranes-1567717 – This paper investigates Efficient phosphorus recovery from municipal wastewater us-ing enhanced biological phosphorus removal and magnesium-based pellets. Few comments below can be considered by authors to improve the manuscript.

- State clearly the research gap and the novelty of the work. P recovery using the propsoed method is rather common and the use of MBR instead of the traditional biological process offer not so strong of a novelty.

- Table 1: prodvide meant and standard deviation of the data.

- "ethanol injection (100 mg/L as COD)" please specify this statement.

- How was the adaptation process during the change of operational parameters between phases. The time series data are required to show the adaptation process of the biological system.

- Figure 5 is explained in term of EPS which maybe misleading. The main cause of the TMP incerent rate was mainly because of the local mixing (with and without aeration)

- Consider to include "membrane beoreactor" in the title to tally with the journal names.

- Many sentences are too-long, which make them hard to read. Try to improve readability.

Author Response

Thanks for your valuable comments. We revised and improved the quality of our manuscript according to your comments and highlighted using yellow color in the manuscript. 

Reviewer 2 Report

An anaerobic/anoxic/aerobic MBR was operated with MgCO3 pellet column for municipal wastewater treatment and P recovery. The reactor was operated in 7 phases. Ethanol addition enhanced the biological nitrogen removal performance. The MgCO3 column effectively adsorbed P in the MBR effluent under different effluent withdrawal modes. High P adsorption capacities was also obtained at high MBR effluent P concentrations. Placing the membrane module into the anoxic zone resulted in high membrane fouling comparing to these in the aerobic zone.

Generally, the manuscript is well-prepared with most parts clearly presented. A few questions needed to be addressed before the manuscript is considered for publication.

  1. The authors aimed to increasing the biological P and N removal efficiencies. However, ethanol was chosen as an external carbon source. Ethanol is not a well-recognized carbon source which could be effectively used by PAOs. Just wondering why did the authors used ethanol other than acetate or glucose. Please specify.
  2. The authors claimed EBPR in the manuscript. However, no direct evidence was shown to prove that there was indeed EBPR activities. May the authors please provide the anaerobic/anoxic P release and aerobic P uptake activity data if they had any.
  3. The first sentence is a bit confusing “the rate of their demand for fertilizers remains at 3% per year”. Please rephrase.
  4. “Membrane Bioreactors (MBRs) have been widely used in the treatment of municipal and industrial wastewater as membrane separation technology is increasingly becoming an innovation in biological wastewater treatment 23–25 The usage of MBR technology is on the rise nowadays as…”. Please fix punctuations.
  5. Table 1. The quality of the primary effluent varied a lot. May the authors please brief explain why. How does it affect the experiment?
  6. “In Phase 4, the MBR was operated at 8 hr HRT with ethanol injection (100 mg/L as COD) into the anaerobic zone.” Please keep a consistency about the location where ethanol was added.
  7. Please check and confirm the effluent TCOD values in Tables 3, 5 and 6. The values seem to be extremely low. What is the detection limit of the method used for TCOD analysis?
  8. Please specify the number of measurements for the data showed in Tables 3-7. 9. May the authors please elaborate a bit more about the mechanisms of ammonia in the MgCO3 pellet column.

Author Response

(The authors gave the same response as above.)

Round 2

Reviewer 1 Report

The comments have been very well addressed.

Author Response

Thanks for your comment.

Reviewer 2 Report

The authors had revised the manuscript accordingly. The MS is now good for publication. However, the author should proofread the ref. list. There are errors. e.g., Oehmen, A.; Lemos, P. C.; Carvalho, G.; Yuan, Z.; Keller, J.; Blackall, L. L.; Reis, M. A. M. Advances in Enhanced Biological 450
Phosphorus Removal: From Micro to Macro Scale. Water Resour. 2007, 41 (11), 2271–2300.

Author Response

Thanks for your comment. The reference list has been updated.